# Identification of biomarkers associated with inflammatory response in Parkinson's disease by bioinformatics and machine learning

Yatan Li[1☉], Wei Jia[2☉], Chen Chen[3], Cheng Chen[4], Jinchao Chen[3], Xinling Yang[5,6]*, Pei Liu[3]*

1 Department of Pharmacy, the Second Affiliated Hospital of Xinjiang Medical University, Urumqi, China, 2 The Affiliated Tumor Hospital of Xinjiang Medical University, Urumqi, China, 3 College of Information Science and Engineering, Xinjiang University, Urumqi, China, 4 College of Software, Xinjiang University, Urumqi, China, 5 Xinjiang Key Laboratory of Neurology, Urumqi, China, 6 Xinjiang Medical University, Urumqi, China

☉ These authors contributed equally to this work and should be considered co-first authors.
* yangxinling2014@163.com (XY); lp374187120@163.com (PL)

## Abstract

Parkinson's disease (PD) is a common and debilitating neurodegenerative disorder. The inflammatory response is essential in the pathogenesis and progression of PD. The goal of this study is to combine bioinformatics and machine learning to screen for biomarker genes related to the inflammatory response in PD. First, differentially expressed genes associated with inflammatory response were screened, PPI networks were constructed and enriched for analysis. LASSO, SVM-RFE and Random Forest algorithms were used to screen biomarker genes. Then, ROC curves were drawn and PD risk predicting models were constructed on the basis of the biomarker genes. Finally, drug sensitivity analysis, mRNA-miRNA network construction and single-cell transcriptome data analysis were performed. The experimental results showed that we screened 31 differentially expressed genes related to inflammatory response. Signaling pathways such as cytokine activity were associated with these genes. Three biomarkers were identified using machine learning algorithms: IL18R1, NMUR1 and RELA. Seventeen co-associated miRNAs were identified by the mRNA-miRNA network as possible regulatory nodes in PD. Finally, these three biomarkers were found to be closely associated with T cells, Endothelial cells, excitatory neurons, inhibitory neurons, and other cells in single-cell transcriptomic analysis. In conclusion, IL18R1, NMUR1 and RELA could be potential therapeutic targets for PD in inflammatory response and new biomarkers for PD diagnosis.

## 1. Introduction

PD is an intractable and neurodegenerative disease associated with high morbidity and disability [1]. There is currently no effective treatment in the medical

**Data availability statement:** The datasets generated and/or analyzed during the current study are available in the Gene Expression Omnibus (GEO) repository, under the accession number GSE20141、GSE20164 and GSE202210. The repository contains all relevant raw data needed to replicate the results of the study, including values for statistical analysis.

**Funding:** This study was funded by the Project about Parkinson's disease peripheral neuropathy drug intervention fusion research innovation team [2023D14006 to Yatan Li]. The funders participated in the study design, data collection and analysis, decision to publish, or manuscript writing.

**Competing interests:** The authors have declared that no competing interests exist.

community, and existing clinical treatments are ineffective [2]. The prevalence of PD is projected to continue to increase over the next 30 years, placing a significant financial and health care burden on society [3]. PD has a set of well-defined features, such as bradykinesia with resting tremor, as well as nonmotor symptoms including autonomic dysfunction, incontinence, and cognitive impairment [4]. When clinical signs of PD first appear, patients lose about 60% of striatal dopaminergic nerve endings and about 30% of nigrostriatal neurons [5]. Because the clinical features of Parkinson's disease are similar to those of other neurodegenerative diseases, and because there are no biomarkers for early and definitive diagnosis, misdiagnosis occurs even when the neurologist makes a diagnosis based on the patient's clinical manifestations and medication response. In addition, studies have shown that PD patients may experience these symptoms decades before diagnosis, resulting in patients not receiving timely treatment [6]. Therefore, the identification of more reliable and sensitive biomarkers is critical for early diagnosis and intervention in PD [7].

The initial manifestation of PD is an inflammatory response, and specific analysis of inflammation-associated genes may reveal useful biomarkers for monitoring disease progression [8]. Microglia-mediated neuroinflammation is thought to be an important aggravating factor driving dopaminergic neuron degeneration and central spread of pathogenic proteins. By exacerbating neuronal damage, neuroinflammation leads to neurodegeneration in PD [9]. Studying the underlying inflammatory mechanisms in PD may help to improve our understanding of disease progression. It may also aid in the development of immunomodulatory drugs and immunotherapeutic strategies to suppress inflammation. The neuroinflammation associated with the pathogenesis of PD has been widely studied. For example, activated microglia secrete a variety of pro-inflammatory cytokines, including tumor necrosis factor-α (TNF-α), interleukin (IL)-1β, and interleukin IL-6 [10–12]. Heidari, A., et al. [13] concluded that the importance of neuroinflammation in PD patients is manifested by elevated serum and cerebrospinal fluid levels of pro-inflammatory cytokines and the appearance of active microglial cells in various regions of the CNS such as the substantia nigra. Araújo, B., et al. [14] showed that microglia remain hyperactivated over time, accompanied by a dysregulated secretion of pro-inflammatory cytokines, thus rendering microglia facilitators or contributing to the neuropathologic and toxic processes of PD. Lv, Q.-K., et al. [15] discussed the interplay of the microglia and α-syn in PD development and the possible mechanisms of microglial autoaggregation and phase phagocytosis in removing α-syn and suppressing neuroinflammation. Yi, M., et al. [16] found that several inflammatory genes are associated with PD risk, including genes related to neurotoxins, metabolism-related enzymes, and genes encoding cytokines. In summary, neuroinflammatory responses are highly implicated in the development and progression of PD. Analysis of inflammatory genes may reveal new PD markers and ideas for further understanding of PD pathogenesis.

In recent years, researchers have combined bioinformatics and machine learning to analyze data from samples such as blood and postmortem substantia nigra (SN)

from PD patients to search for potential biomarkers of PD and to assist clinicians in early diagnosis and detection of PD patients [17–20]. For example, Lei, C., et al. [17] searched for PD necrosis-associated genes by comprehensive cluster analysis, enrichment analysis, and WGCNA analysis. Yang, X. and Z. Wang [18] searched for new immune-related biomarkers of Parkinson's disease by developing a PD diagnostic model using Lasso and multivariate Cox regression. Chen, X., et al. [19] Search for potential biomarkers of Parkinson's disease using gene ontology (GO) enrichment and gene set variation analysis (GSVA). Bao, Y., et al. [20] screened genes closely related to PD using LASSO and SVM algorithms, which provided new perspectives for further research on PD immune mechanisms and treatment. In summary, the combination of bioinformatics technology and machine learning is expected to accurately identify potential biomarkers of PD, which is important for early detection of PD patients, patient stratification, and disease progression monitoring.

In this study, we downloaded the GSE20141 and GSE20164 datasets containing normal and PD samples from the gene expression omnibus(GEO) database platform [21]. Differential expression of inflammatory response genes was screened using the GSE20141 dataset. PPI networks, GO and KEGG pathway analysis were conducted for the differential expression genes using bioinformatics tools. Then Lasso, SVM-RFE and RSF machine learning algorithms were used to get candidate genes from these differential genes, and the intersection was taken to get the biomarkers for PD. Validate the diagnostic performance of biomarkers using ROC curves. Corresponding PD risk prediction models were also constructed to explore the predictive ability of these biomarkers. And the validity of these biomarkers was validated in the external dataset GSE20164. Subsequently, the biomarkers were subjected to drug-sensitive analysis and corresponding mRNA-miRNA regulatory network construction, which provided insight into the biomarkers' biological functions and potential mechanisms. Finally, single-cell transcriptomic analysis revealed that the screened biomarkers were abundantly expressed in different cell types. The entire analysis procedure is shown in Fig 1.

## 2. Meterials and methods

### 2.1 Data collection

In this experiment, we obtained two PD-related datasets, GSE20141 and GSE20164, on GEO [22]. The biotypes for these datasets are all Homo sapiens and the sample tissues are from the substantia nigra of the brain. The GSE20141 database was used as the main study object, and the expression profiling data generated using the GPL570 data platform included 18 samples, of which 8 and 8 were in normal and PD group, respectfully. The GSE20164 database served as the external validation subject, and the GPL96 data platform was used to generate the expression profiling data, which contained 11 samples, of which 5 and 6 were in the control and PD groups, respectively. Two hundred genes related to the immune response were downloaded from the human gene set HALLMARK_IFLAMMATORY_RESPONSE by the Gene Set Enrichment Analysis (GSEA) platform [23].

### 2.2 Inflammation-related differential gene screening

Microarray data from the GSE20141 dataset were used to screen for differential gene expression between controls and PD groups with the R "limma" package [24]. Screening conditions for differential genes were $P < 0.05$, $|log2FC| > 1$. Heatmaps and volcano plots were used to illustrate these differential genes. The screened differential genes were taken to intersect with genes related to the inflammatory reaction.

### 2.3 Immune cell infiltration analysis

The proportion of 22 autoimmune tumor-infiltrating cell types in PD was evaluated by reverse convolution algorithm (CIBERSORT) [25]. Differential expression maps and correlation heat maps of immune cell components in the normal and PD groups were graphed with the appropriate R packages.

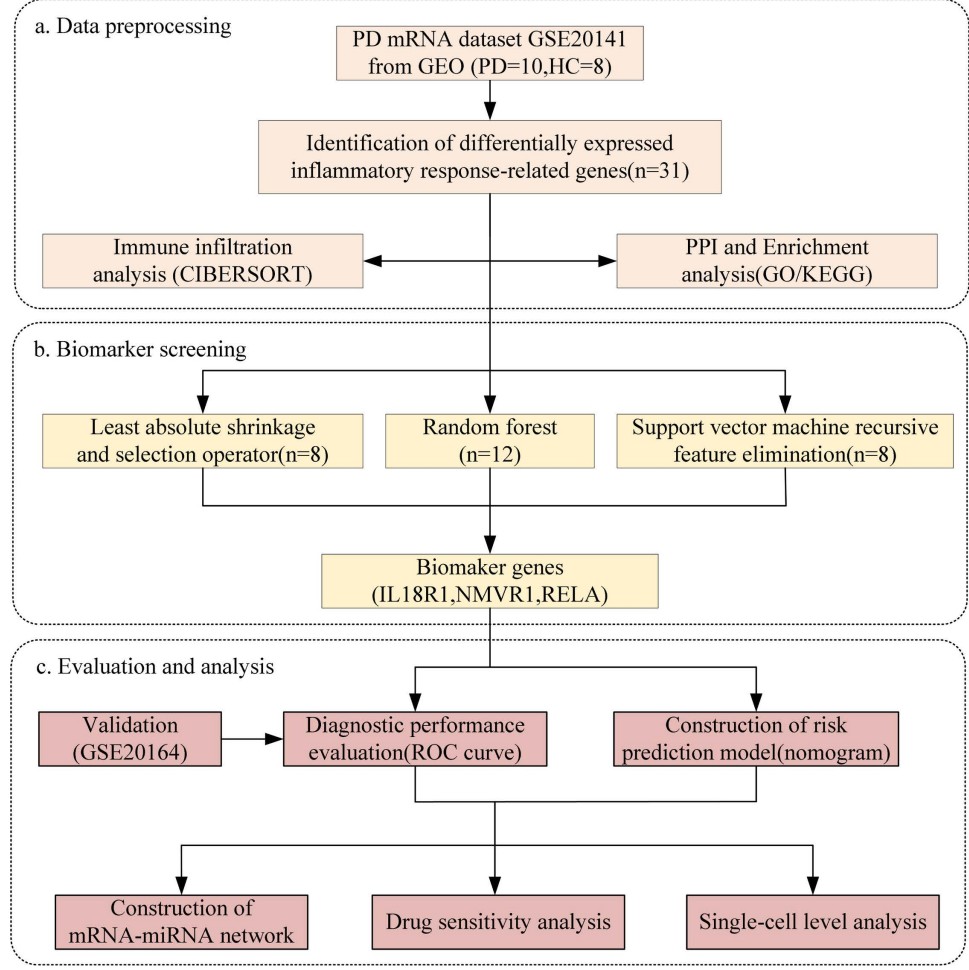

**Fig 1. General workflow of this study.**

## 2.4 PPI network building and differential gene enrichment analysis

The differentially identified genes were input into the STRING online tool to obtain protein-protein interaction network (PPI) data, with the minimum interaction value set at 0.4. The PPI network of the screened differential genes was visualized in Cytoscape software [26]. Finally, differentially expressed genes associated with inflammatory response were analyzed for GO enrichment [27] and KEGG pathway enrichment [28], respectively. GO enrichment analysis can understand the common features of the screened differential genes in terms of biological processes, molecular functions, and cellular composition. KEGG pathway enrichment analysis explores the functions and regulatory mechanisms of the screened differential genes in organisms by detecting the enrichment of pathways in these genes.

## 2.5 Multiple machine learning algorithms for biomarker screening

The screened inflammation-related differential genes were further screened using LASSO [29], SVM-RFE [30], and RSF [31]algorithms. The LASSO regression algorithm is suitable for multicollinearity problems and is able to implement a regression method for variable selection along with parameter estimation. The complexity is adjusted by the parameter λ. For linear models with more variables, the penalty is greater for larger parameters. Finally, a model with fewer variables

can be obtained. Here, the optimal value of λ is continuously adjusted using the "glmnet" package of R. A backward selection method is the SVM-RFE algorithm. The RSF algorithm assigns an importance score to each differential gene, and feature screening is achieved by ranking the magnitude of the importance score of each gene. By further screening the previously screened differential genes using each of these machine learning algorithms, three gene sets were obtained. The intersection of these three gene sets is taken to obtain the potential biomarkers for PD.

### 2.6 Assessment of the diagnostic performance of biomarkers

On the GSE20141 and GSE20164 validation sets, we verified that these biomarkers have diagnostic value for PD using the ROC curve.

### 2.7 Biomarker-based PD risk prediction models

Using these screened biomarkers, we developed a biomarker gene-based risk prediction model for PD. The reliability of the risk predictive model for these biomarkers was verified by constructing a nomogram of PD patients, ROC curves for patient classification, calibration curves, and clinical curves.

### 2.8 Drug sensitivity analysis of biomarkers

We searched for biomarker-related drug information in the CellMiner database [32]. Finally, the results were visualized using the R package "ggplot2" [33].

### 2.9 Biomarker-based mRNA-miRNA regulatory networks

The biomarker-associated miRNAs were obtained using the online website Mirwalk [34]. This site is a comprehensive database of miRNA-target gene interactions containing information on miRNA target genes from a wide range of species. Interactions of biomarkers with relevant miRNAs are shown. The mRNA-miRNA regulatory network was mapped using the Cytoscape tool [26].

### 2.10 Processing and analyzing single-cell transcriptome data

The initial single-cell transcriptome data for Parkinson's disease were taken from the GSE20228 dataset. During data processing, we utilized the R package Seurat (http://satijalab.org/seurat/) [35] to normalize, scale, and cluster the single-cell data. For quality control of the single cell data, nFeature_RNA<=8000, nCount_RNA<=30000, and percent.mt<=3 were used. Then, different cell types were tagged according to the specifically expressed genes in each cluster. Finally, the "runTSNE" function was used to visualize the distribution of different cell clusters.

## 3. Results

### 3.1 Identify differentially expressed genes related to PD inflammation

The PD dataset GSE20141 was screened for the differentially expressed genes using the method of limma of variance at $P < 0.05$ and |log2FC| > 1. As a result, 2855 differentially expression genes (DEGs) were identified. Among them, 394 were upregulated and 2461 were downregulated. These differentially expressed genes were visualized by heat map (Fig 2A). Then, intersections with 200 inflammation response related genes (IRRGs) were performed to yield 31 inflammation response related differentially expressed genes (Fig 2B). And from the volcano graphs of differentially expressed genes, it was found that 30 of these 31 differential expression genes related to inflammatory response were downregulated genes and 1 was upregulated gene (Fig 2C). Finally, the correlation between these 31 genes was analyzed. It was found that the PCDH7 gene showed negative correlation with most of the genes and positive correlation with some of the genes (Fig 2D).

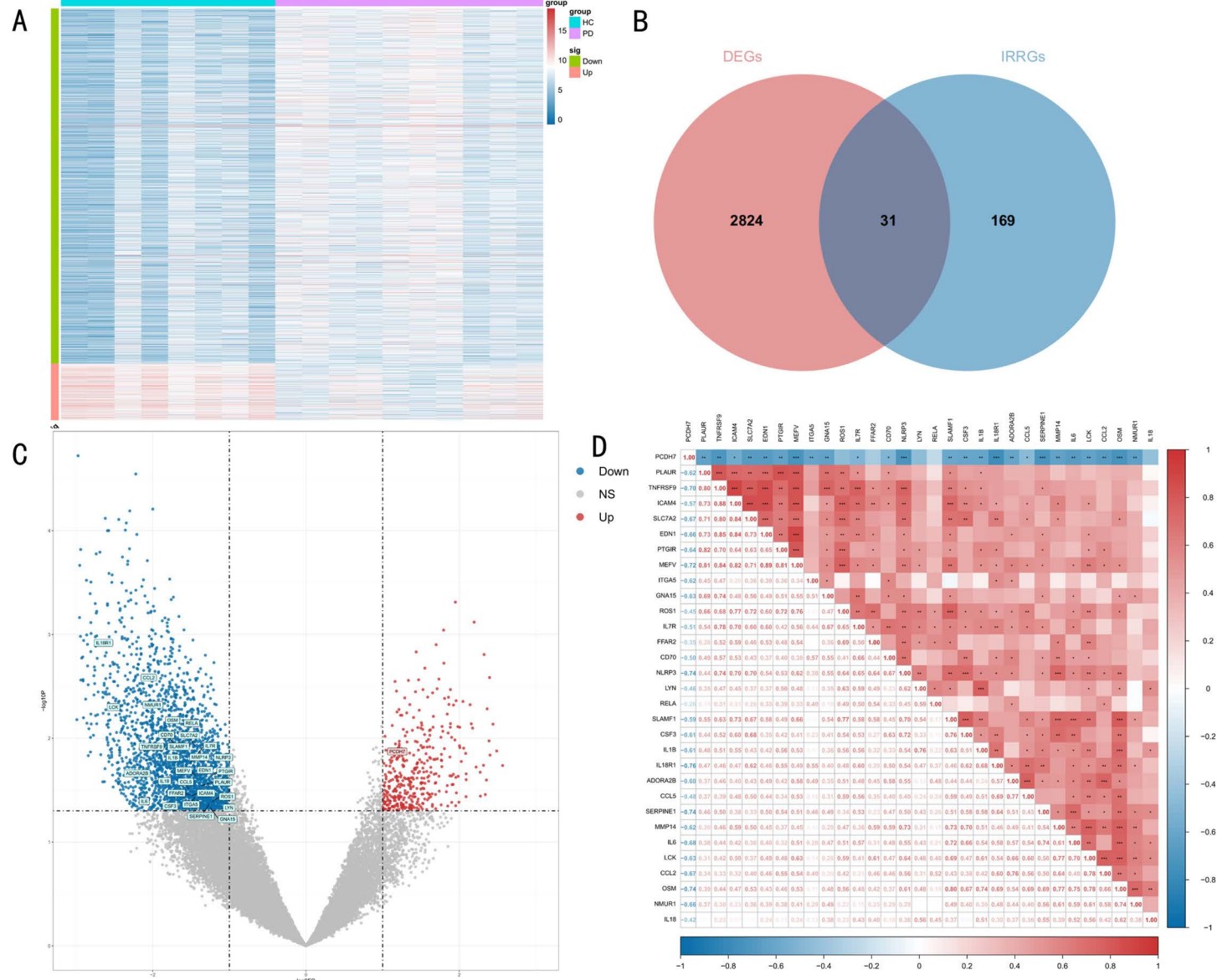

**Fig 2. Results of the differentially regulated expression analysis of inflammatory response genes in PD.** (A) Heat map of differential expression genes in PD. (B) Intersection Venn diagram of differential expression genes and inflammatory response related genes. (C) Volcano diagram of differential expression genes in PD. (D) Correlation diagram of 31 differentially expressed inflammatory response associated genes.

## 3.2 Analysis of the correlation of immune cell infiltration in PD

Further analyses of 31 differently expressing immune response genes examined the relationship between PD and immune-cell infiltrates. First, the proportions of the 22 different immune cells present in all samples of the GSE20141 data-set were calculated using the CIBERSORT algorithm (Fig 3A). The differential expression of immune cell infiltration in PD was then analyzed. The analysis showed differential expression of both activated mast cells and neutrophil in PD (Fig 3B). Then, the correlation between 22 immune cell infiltrates was analyzed again, and the results were shown in the correlation heat map (Fig 3C). Finally, association analysis was performed between differential expression of inflammation response

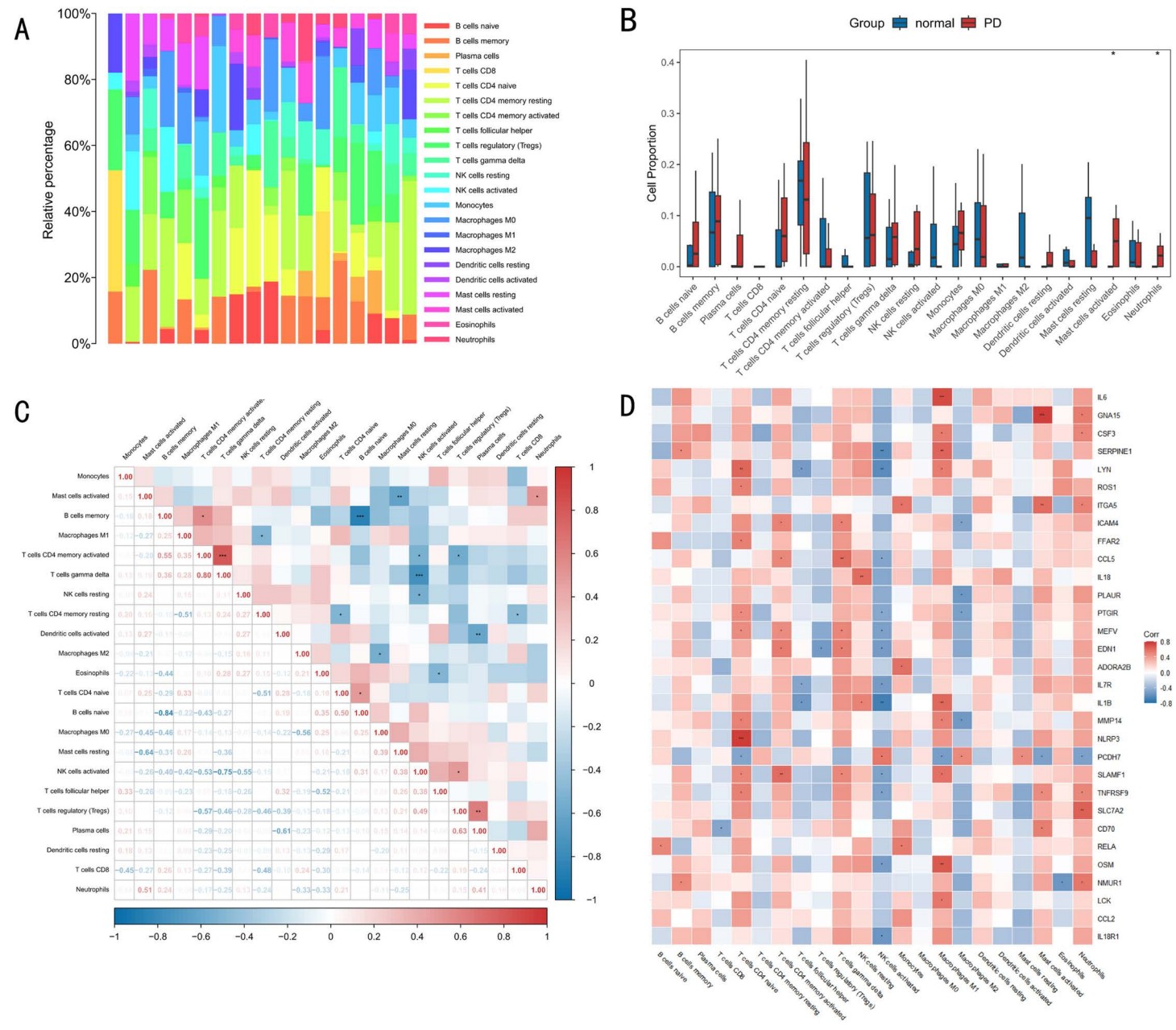

**Fig 3. Results of correlation analysis of the immune cell infiltrates in PD.** (A) Plots of 22 immune cell proportions in all samples of data set GSE20141. (B) Differential expression plot of the immune cell infiltrates. The blue and red colors indicate normal and PD samples, respectively. (C) Correlation plots of 22 immune-cell infiltrations. (D) Construction of association graph between differential expression of inflammatory response-related genes and immune-cell infiltrates. * P < 0.05; ** P < 0.01; *** P < 0.001.

related gene and immune cell infiltration (Fig 3D). The analysis showed a strong association between several differentially expressed inflammatory response related genes and immune cells, such as NLRP3 and CD4 naive T cells (Spearman r = 0.77, p < 0.001), GNA15 and activated mast cells (Spearman r = 0.76, p < 0.001), IL6 and M1 macrophages (Spearman r = 0.72, p < 0.001) showed a strong positive correlation.

## 3.3 Building PPI networks and analyzing GO and KEGG enrichment

In order to further investigate the possible interactions of the 31 differentially expressing inflammation response-related genes in PD, a PPI network was first constructed via the string website, and a sum of 31 nodes and 83 edges was obtained (Fig 4A). The PPI network was redrawn utilizing Cytoscape software. The shading of the gene color indicates the expression level (Fig 4B). To further investigate the function and localization of these 31 differentially expressed

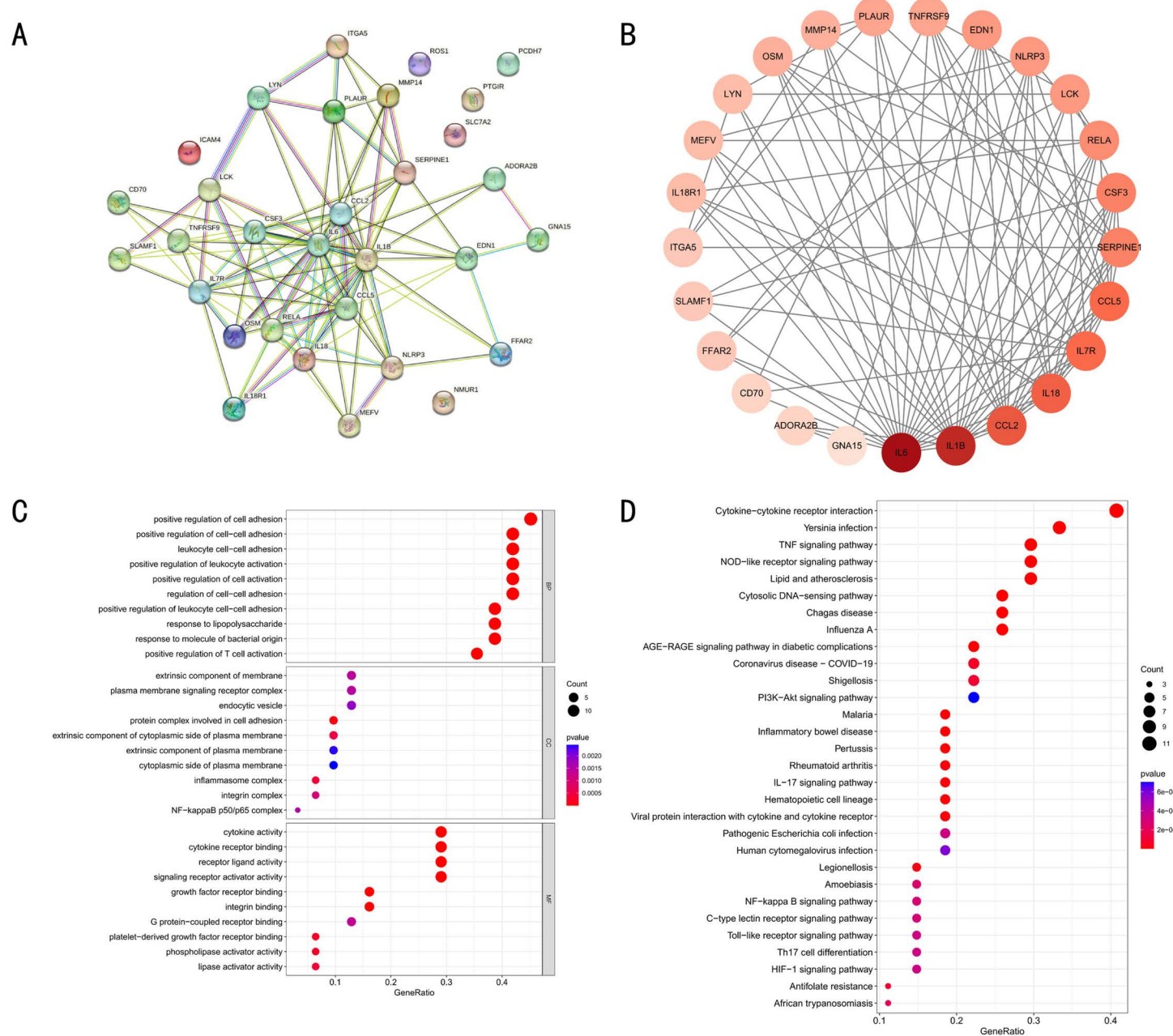

**Fig 4. PPI network and results of GO and KEGG analysis.** (A) PPI network based on 31 differentially expressed genes associated with inflammatory response. (B) PPI network plotted in Cytoscape software. (C) Graph of GO analysis results. (D) Graph of KEGG analysis results.

inflammatory response-related genes, GO and KEGG analyses were conducted. GO enrichment analysis data indicated that most of these genes were associated with positive regulation of cell adhesion in the BP class and positive regulation of cell adhesion, etc., with extramembrane components and plasma membrane signaling receptor complexes, etc. in the CC class, and with cytokine activity and cytokine receptor binding, etc. in the MF class (Fig 4C). In addition, KEGG pathway analysis revealed that most of the genes were associated with cytokine-cytokine receptor interaction, Yersinia infection, and TNF signaling pathway (Fig 4D).

### 3.4 Machine learning-based biomarker screening

Three different machine learning algorithms were then applied to further screen the candidate genes. The 8-fold cross-validated LASSO algorithm was used first, and a total of eight candidate genes were screened (Fig 5A). They were IL18R1, NMUR1, RELA, CD70, SLC7A2, PTGIR, LYN, and GNA15. Next, 8-fold cross-validation was performed on the SVM-RFE algorithm. The false discovery rate was minimal when the candidate gene number was 8 (Fig 5B). They were RELA, CD70, NMUR1, SLAMF1, ADORA2B, IL18R1, CCL2, and IL6, respectively, and then the RandomForest algorithm was used to obtain the gene importance score ranking (Fig 5C). This RandomForest experiment was set to screen for candidate genes with MeanDecreaseGini>0.3. Thus, the number of candidate genes screened was 12, which were NMUR1, IL18R1, LCK, PTGIR, PCDH7, RELA, ITGA5, OSM, SLC7A2, EDN1, CCL2, and FFAR2. Finally, the candidate genes obtained from these algorithms are taken to intersect to obtain 3 biomarkers. The three biomarker biomarker genes were IL18R1, NMUR1, and RELA (Fig 5D).

### 3.5 Diagnostic performance assessment of biomarker genes

To further explore the potential diagnostic and prognostic properties of these three biomarkers in PD, this experiment first examined their expression in PD. The results showed that all three biomarker genes were expressed higher than normal in PD patients (Fig 6A). The diagnostic performance of three biomarkers, IL18R1, NMUR1 and RELA, was then evaluated. The ROC curves showed that IL18R1 (AUC:0.913), NMUR1 (AUC:0.838) and RELA (AUC:0.825) all had good diagnostic performance, suggesting that they may be potential therapeutic targets for PD (Fig 6B). The next step was to verify whether IL18R1, NMUR1, and RELA had generalization ability and robustness. This work was further validated on an external dataset, GSE20164, and found that NMUR1 and RELA were differentially expressed, while the IL18R1 differential expression profile was not significant (Fig 6C). Finally, the ROC curves showed that IL18R1 (AUC:0.633), NMUR1 (AUC:0.900), and RELA (AUC:0.867) also had good diagnostic performance on the validation set GSE20164(Fig 6D).

### 3.6 Construction of a biomarker-based risk prediction model for PD

To further explore the risk predictive ability and clinical value of IL18R1, NMUR1 and RELA, this study integrated these three biomarkers and constructed a PD risk prediction model (Fig 7A). A score is obtained based on the high and low levels of expression of each biomarker gene in the nomogram, and the total score corresponds to its risk value. For example, patients with low expression of the RELA gene, high NMUR1 gene expression, and low IL18R1 gene expression would receive a total score of approximately 196 points, corresponding to a risk prediction probability of 0.5. Then, the ROC curve was plotted and the AUC of this model was 0.968, which indicates that the model has a strong prediction ability (Fig 7B). The calibration curve revealed a small difference between the real and the predicted PD risk, indicating the high prediction accuracy of the model (Fig 7C). Finally, the clinical impact curve showed that the model was more similar to the actual situation in clinical assessment (Fig 7D). Finally, it was verified that IL18R1, NMUR1 and RELA could be potential targets for PD prediction analysis with prognostic ability.

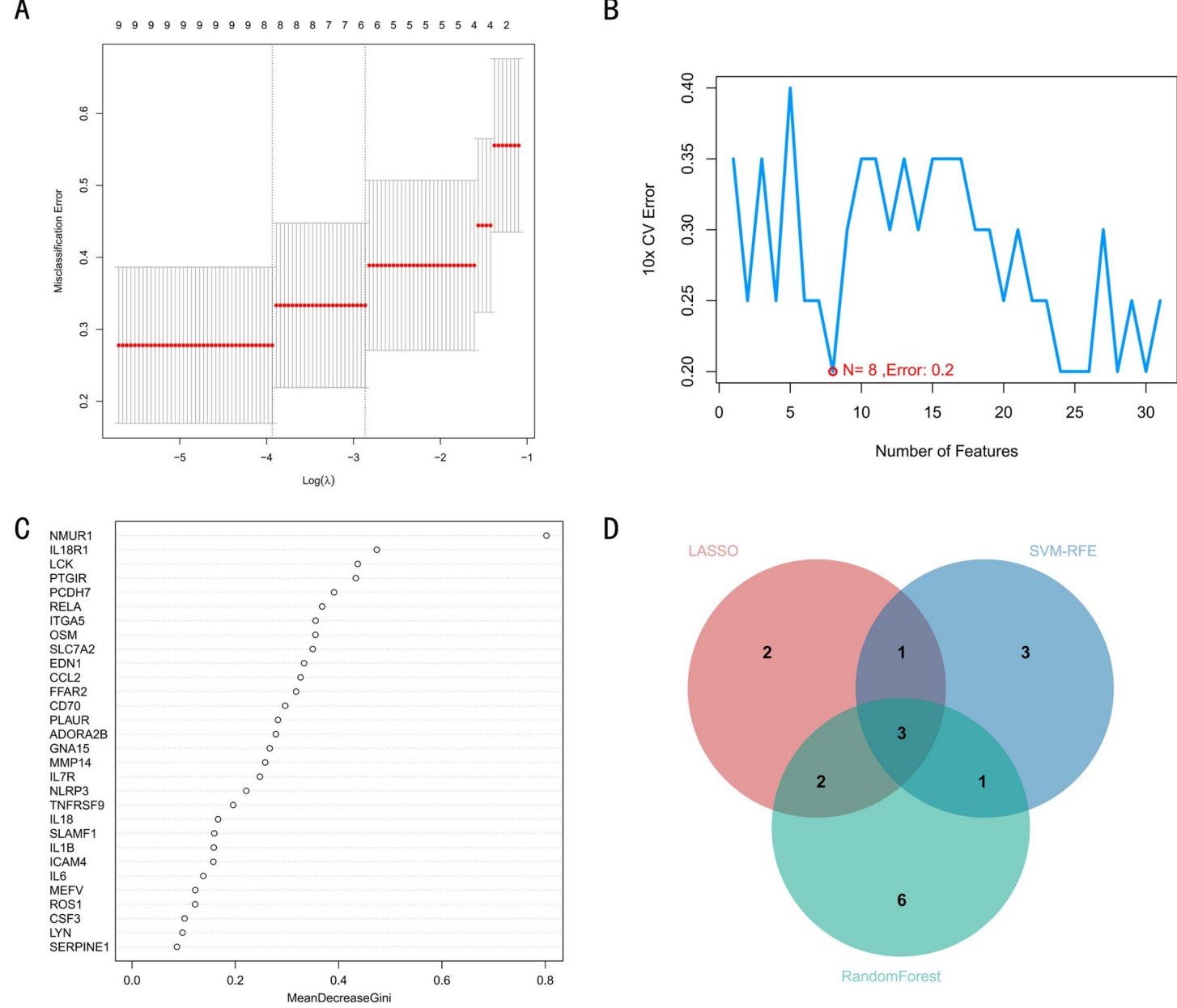

**Fig 5. Results of machine learning based biomarker screening analysis.** (A) Feature selection graph of LASSO algorithm. (B) Error rate graph of SVM-RFE algorithm. The red dots indicate the numbers of candidate candidate genes with the minimal false discovery rate. (C) Sorting graph of gene importance scores for the RandomForest algorithm. (D) Venn diagram of the intersection of the gene candidates examined by the three machine learning algorithms.

## 3.7 Drug sensitivity analysis of biomarkers

Drug sensitivity analysis helps identify potential drug targets for Parkinson's patients and generates new ideas for treating Parkinson's patients. The CellMiner database was used to correlate the drug sensitivity of IL18R1, NMUR1 and RELA. In this paper, $P < 0.05$ and $|Cor| > 0.3$ were used as screening criteria. IL18R1 was positively correlated with the

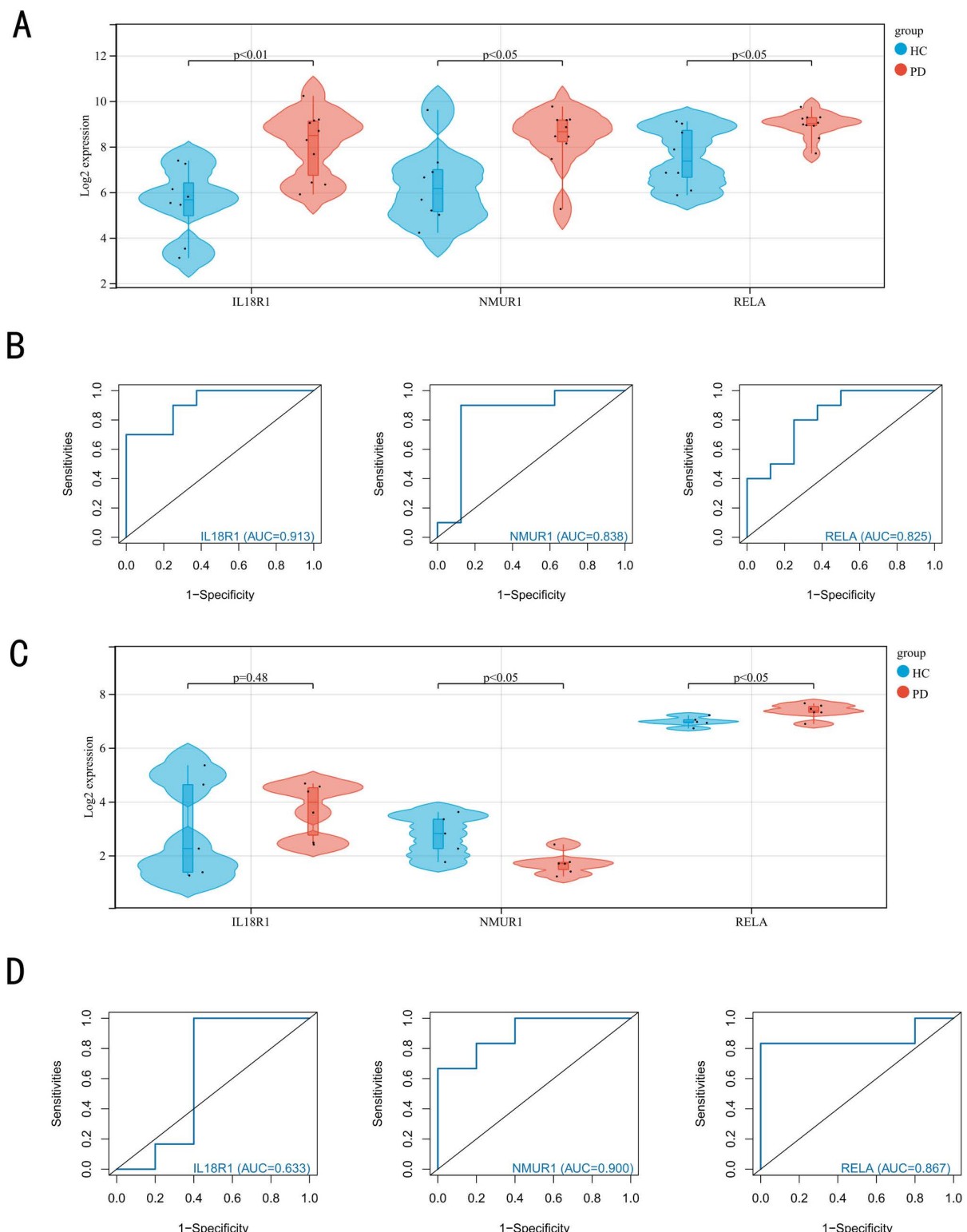

**Fig 6. Expression and diagnostic performance analysis results of IL18R1, NMUR1 and RELA genes.** (A) Expression of biomarker genes in GSE20141 dataset. (B) Evaluation of biomarkers with ROC curves. (C) Expression of biomarker genes on validation set GSE20164. (D) ROC curve evaluating the diagnostic performance of biomarker genes on validation set GSE20164.

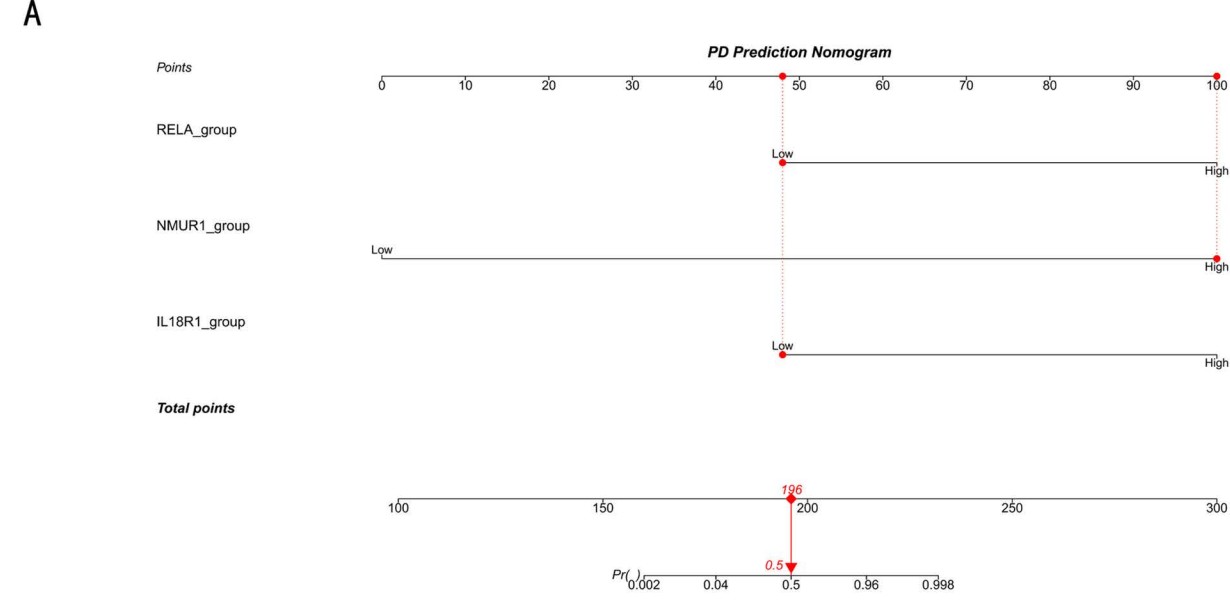

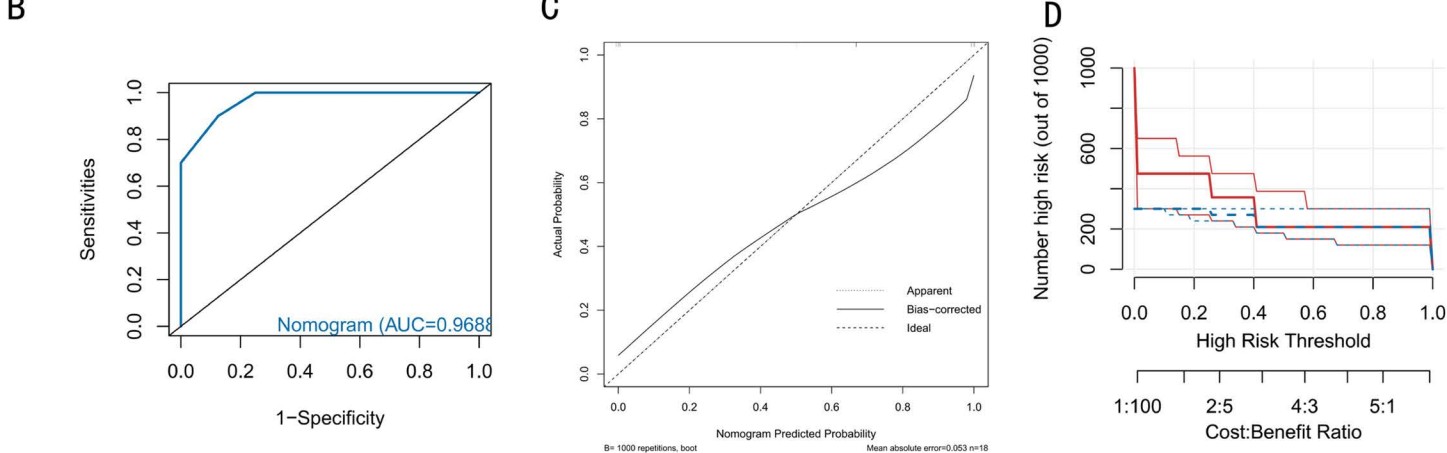

**Fig 7. Construction of PD risk predicting model and evaluation indices.** (A) Construction of the nomogram of the PD disease risk prognosis model on the basis of the three biomarkers. (B) ROC curve of the model. (C) The calibration curve evaluating the predictive accuracy of the model. (D) Clinical impact curve evaluating the model's clinical performance.

drugs Alisertib (Cor = 0.31, P = 0.016), Azacitidine (Cor = 0.33, P = 0.01) and Motesanib (Cor = 0.35, P = 0.007), IL18R1 was negatively correlated with the drug BP-1–82 (Cor = - 0.36, P = 0.004), NMUR1 was negatively correlated with Lex-ibulin (Cor = 0.31, P = 0. 017) and Norvir (Cor = 0.48, P < 0.001), RELA was positively correlated with CAMPTOTHECIN (Cor = 0.32, P = 0.012) and Clofarbine (Cor = 0.3, P = 0.019) and 7 other drugs, and RELA was positively correlated with ARRY-162 (Cor = -0.37, P = 0.004) and Tamoxifen (Cor = -0.3, P = 0.018) and 7 other drugs (Fig 8).

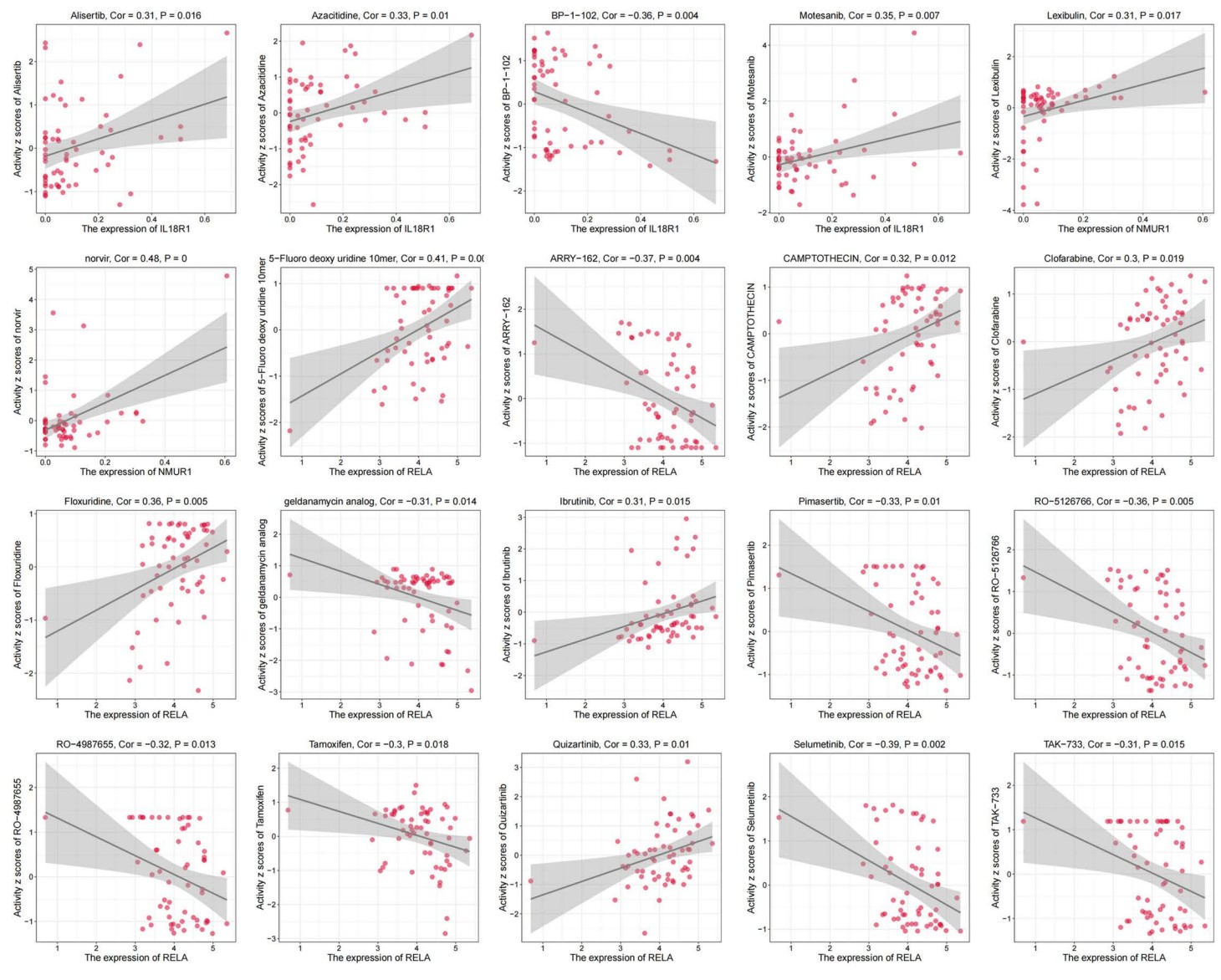

**Fig 8. Results of drug sensitivity correlation analysis based on IL18R1, NMUR1 andRELA genes.**

### 3.8 Construction of a biomarker-based mRNA-miRNA network

To further investigate the upstream regulatory networks and potential molecular mechanisms of the three biomarker genes IL18R1, NMUR1 and RELA. In this paper, we queried the miRNAs upstream of the three biomarker gene via the Mirwalk online database, and the results showed that there were 427 miRNAs associated with IL18R1 gene, 325 miRNAs associated with NMUR1 gene, and 374 miRNAs associated with RELA gene. And Venn diagram was drawn to screen 17 miRNAs with common predictive relationships of the three biomarker genes (Fig 9A). Then, the mRNA-miRNA regulatory networks of the three biomarker genes IL18R1, NMUR1 and RELA with the 17 miRNAs were mapped using Cytoscape software (Fig 9B). We further explored the roles of these 17 miRNA target genes in the relevant pathways (S1 Fig). Through KEGG and GO analysis of miRNA target genes, we found that the pathways of miRNA target genes overlapped

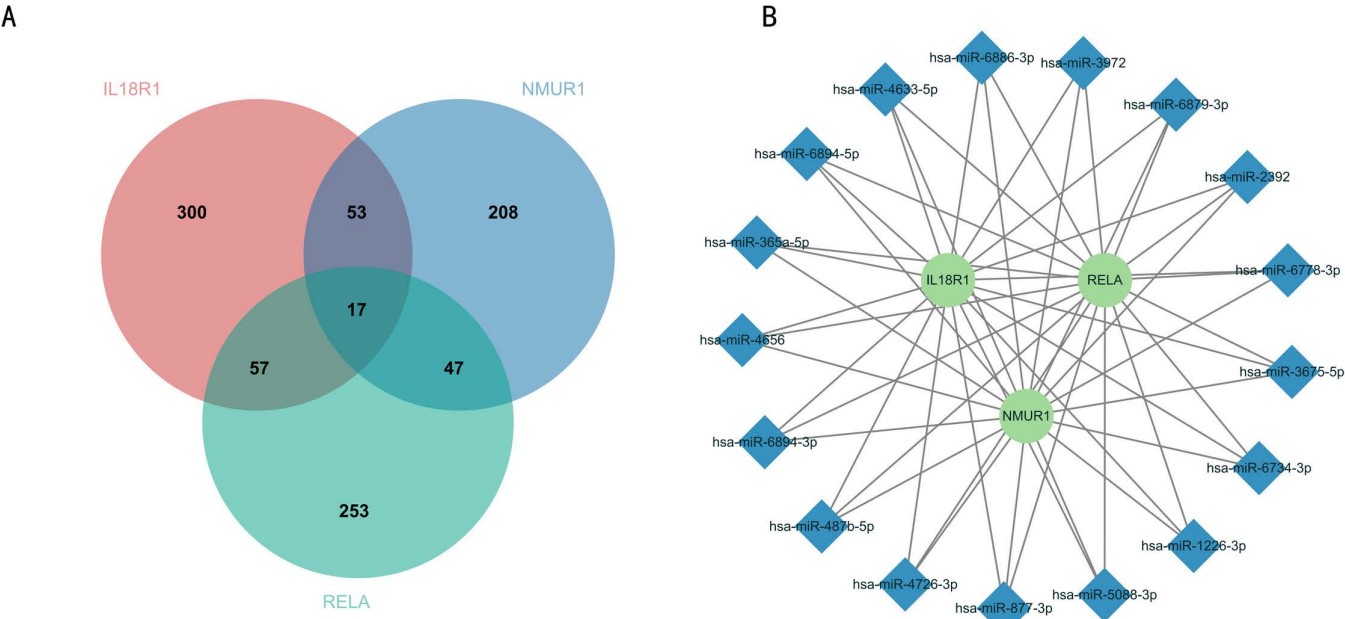

**Fig 9. Results of mRNA-miRNA network construction based on IL18R1, NMUR1 and RELA biomarkers.** (A) The Venn diagram shows the miRNAs associated with IL18R1, NMUR1, and RELA.(B) Mapping of the mRNA-miRNA regulatory network of IL18R1, NMUR1, and RELA.

with those at the mRNA level. These pathways, although significant (P<0.05), were ranked low and therefore did not appear in the pathway heatmap for mRNAs. Specifically, these include the virus process and ECM-receptor interaction pathways. The final results indicated that these 17 miRNAs could be potential regulatory targets in PD, and the network constructed with IL18R1, NMUR1 and RELA biomarkers was one of the important regulatory networks in PD.

### 3.9 Single-cell transcriptomic data biomarker expression

Clustering and manually annotating the raw single-cell transcriptome data yielded eight major cell types (Fig 10A). All of these cells are intimately involved in the pathogenesis of PD. One example is microglia, the most abundant immune cell in the central nervous system, which have been linked to the progression of PD. And to further understand the relative proportions of specific cells in the total cells, we plotted stacked bars for each sample (Fig 10B). In further detail, we displayed the expression of IL18R1, NMUR1, and RELA in these cell populations in bubble plots (Fig 10C) and feature plots (Fig 10D) to further reveal the role of these biomarkers in the progression of PD. The results indicated that RELA was abundantly expressed in these cell populations. IL18R1 and NMUR1 were abundantly expressed in endothelial cells and inhibitory neurons, respectively, and were closely associated with other cell populations.

### 4. Discussion

PD is a progressive neurological disease characterized by the death of midbrain dopaminergic neurons, α-synuclein aggregates, and movement disorders [36]. A major factor in the decline of dopaminergic neurons is the neuroinflammatory response [37]. Because PD is directly linked to genetics, single nucleotide polymorphisms (SNPs) in immune and inflammatory genes may increase the risks of developing PD in people with PD by affecting the body's immune system and inflammation response [16].

In this work, we initially identify 2855 differently represented genes in the GSE20141 dataset. We used bioinformatics tools to do this. Then, 31 inflammatory response-related differently expressing genes were obtained by intersecting with 200

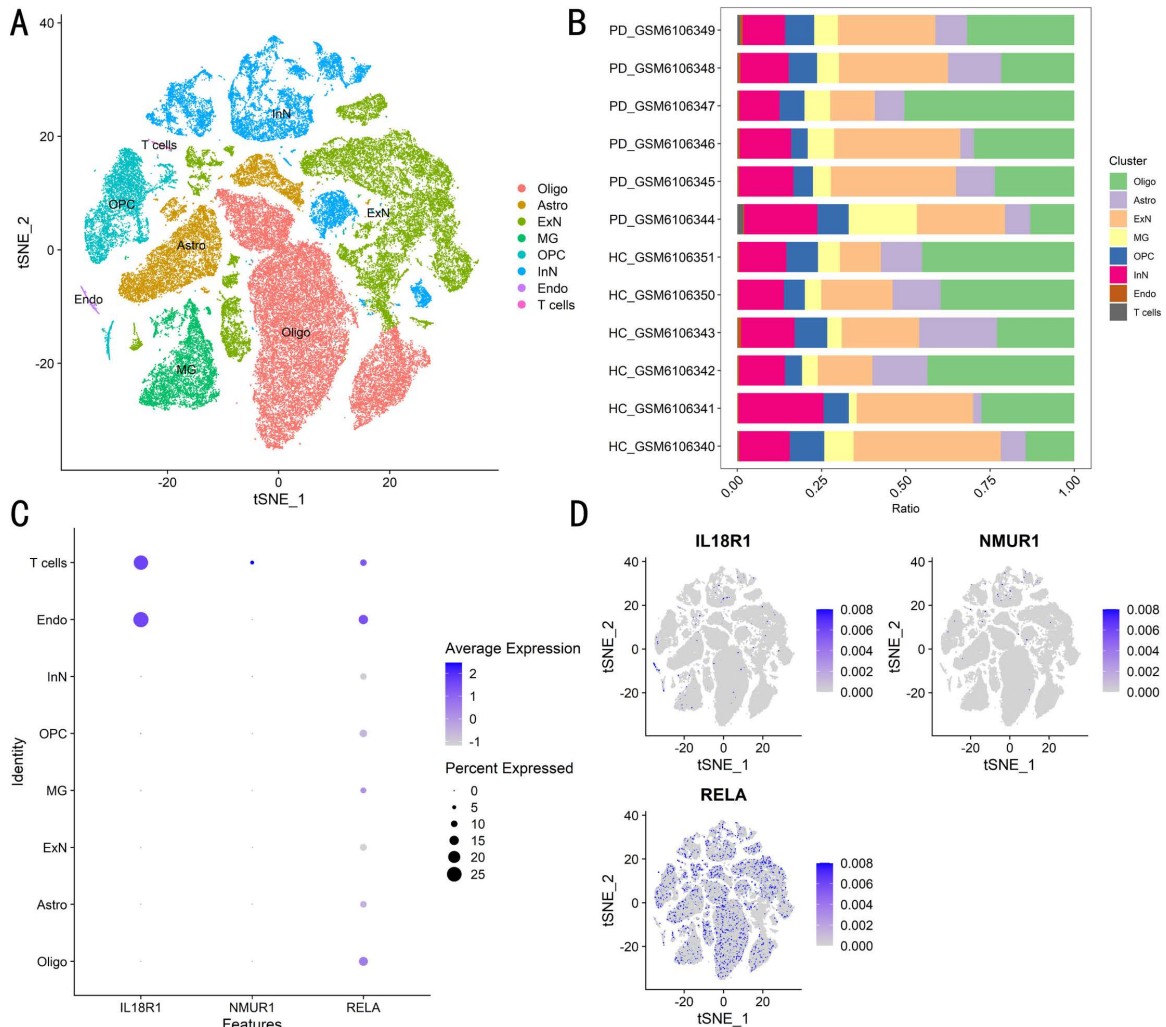

**Fig 10. Analysis of IL18R1, NMUR1 and RELA based single cell transcriptome data.** (A) The 8 major cell types detected in the single cell transcriptome dataset GSE20228. (B) The proportions of different cell types in the total cells. (C) Bubble plots of IL18R1, NMUR1 and RELA expression in these cell populations. (D) Characterization plots of IL18R1, NMUR1 and RELA expression in these cell populations.

human inflammatory response corresponding genes. And these 31 inflammatory response-related differentially expressed genes were evaluated by PPI network, GO analysis and KEGG pathway analysis. Second, LASSO, SVM-REF and RSF machine learning algorithms were employed to further screen the features, and three biomarkers, IL18R1, NMUR1 and RELA, were obtained, and these three biomarkers were validated in the external dataset GSE20164. IL18R1, NMUR1 and RELA were subjected to drug sensitivity analysis and these three biomarkers were found to be closely associated with several drug components. Specifically, IL18R1 was negatively correlated with the drug BP-1–102 (Cor = -0.36, P = 0.004), and RELA was negatively correlated with the drugs ARRY-162 (Cor = -0.37, P = 0.004) and tamoxifen (Cor = -0.3, P = 0.018). The negative correlation implies that the effect of the drug tends to diminish when the expression level of the gene is elevated. For example, Xu, F., et al. [38] found a negative correlation between the expression of SLC1A5 and the sensitivity to camptothecin-based drugs in their analysis of drug sensitivity to classical chemotherapeutic agents. Additionally, because this study is in the preliminary exploratory stage, drugs with weak correlations may still uncover potential complex biological

mechanisms. The relationship between gene expression and drug response is often nonlinear, so a single weak correlation does not necessarily indicate a lack of biological significance. For example, Song, D., et al. [39] found that the drug Sapitinib, which has a weak correlation with NRAV expression, can induce apoptosis and inhibit the phosphorylation of the epidermal growth factor receptor (EGFR) and its downstream pathways when investigating the sensitivity of small-molecule drugs related to NRAV expression. 17 co-associated miRNAs were identified by the mRNA-miRNA network, which may be one of the regulatory nodes in PD. In single-cell transcriptomic analysis, the three biomarkers screened were abundantly expressed in different cell populations. IL18R1 was specifically expressed in T cells and Endothelial cells. It suggests that IL18R1 may play an important role in the immune response, especially in T cells. T cells are key cells of the immune system and are involved in recognizing and attacking infected or tumor cells [40]. Whereas IL18R1 is a subunit of the IL-18 receptor, IL-18 is a pro-inflammatory cytokine that is commonly associated with immune response and inflammation [41]. RELA is expressed in almost all cell populations. The expression of RELA in virtually all cell populations suggests that it is a fundamental and broadly regulated factor that may maintain essential functions and responsiveness in the physiological state of the cell, especially in response to inflammation, infection, or cellular damage [42]. NMUR1 is specifically expressed in T cells. NMUR1 is usually associated with neuropeptide signaling, which may influence the immune response by regulating T cell activation, proliferation, or function [43]. It is worth noting that although the three biomarkers we screened showed extremely strong correlations with PD in ROC curves, nomograms, and various types of bioconfidence analyses on external datasets. However, all current related studies only indirectly indicate that these three biomarkers and PD have an association. Therefore, as newly discovered biomarkers for PD, their reliability warrants further investigation.

The protein encoded by IL18R1 is a member of the interleukin-1 (IL-1) receptor family of cytokine receptors [44]. IL18R1 binds specific to interleukin 18 (IL18) and has an important function in IL18-mediated signaling. Prior studies have demonstrated that IL-18 is a key mediator of neuroinflammation and neurodegeneration in the central nervous system under pathological conditions [45]. The most prominent PPI hotspot identified by Henderson, A.R., et al. [46] in their analysis was the IL18/IL18R1 pathway, which clearly demonstrated the up-regulatory effect of hypomethylation on IL18/IL18R1 expression and highlighted the contribution of IL18 to the neuroinflammatory process in PD. In summary, the IL18R1 biomarker screened acts as an inflammatory gene that may influence the progression of Parkinson's disease.

Neuromedin U contains 2 receptors (NMUR1 and NMUR2) that are distributed throughout the nervous system and many tissues and organs, and has a variety of biological roles [47]. NMUR1 and NMUR2 have multiple physiological functions in the peripheral tissue and the central nervous system (CNS) [48]. The transcription factor nuclear factor-kB (NF-kB), a key regulator of inflammation and apoptosis, has been implicated in the pathology of various neural degenerative diseases [49]. The NF-kB family is made up of five distinct subunits that bind to each other to form transcriptionally active dimers [50]. Nigrostriatal dopamine (DA) neurons in patients with PD have abnormally increased nuclear levels of RELA (NF-KB subunit) [51]. NF-kB is implicated in processes such as inflammation and immunity and acts as a pleiotropic transcription factor. Choi, D.C., et al. [52] found that NF-kB inhibition, but not its activity, is the underlying cell death mechanisms in MPP+ toxicity. Lanzillotta, A., et al. [53] ound that the acetylated state of RELA subunits promotes both neuroprotection and neurotoxicity. Overall, disease progression in PD patients is accompanied by the development of an inflammatory response, and there is a potential relationship between the inflammatory gene RELA and the pathogenesis of PD patients.

In conclusion, three biomarker genes, IL18R1, NMUR1 and RELA, identified in this paper are strongly associated with PD. These three genes may be potential therapeutic targets for PD with prognostic predictive value and clinical value as potential biomarker genes for PD.

## 5. Conclusion

Although the three biomarker genes we investigated showed high correlation with PD in ROC curve analysis on outside datasets, drug susceptibility, mRNA-miRNA regulatory networks, single-cell transcriptome data analysis, and related

research reports. However, the specific roles of the three identified biomarkers in PD need to be elucidated by further clinical and animal studies. However, drug sensitivity analyses were not performed on single-cell transcriptomics data as the primary objective of this study was to validate the specific expression of marker genes in different cell types rather than to explore the direct effects of drugs on these cell types. However, in the future, we will focus on drug sensitivity analysis of single-cell transcriptomics data.

In conclusion, in this study we identified three PD-related inflammatory response genes IL18R1, NMUR1 and RELA using a combination of several bioinformatics methods and several machine learning algorithms. These biomarkers have an enormous impact on the monitoring of PD progression. They can help clinicians detect the progression of PD earlier, enable more accurate diagnosis, and improve the prognosis of PD patients.

## Supporting information

**S1 Fig. S1 Fig KEGG and GO analysis of miRNA target genes.**
(TIF)

**S1 File. Data Availability Statement.**
(DOCX)

## Author contributions

**Conceptualization:** Yatan Li, Wei Jia, Chen Chen, Cheng Chen, Jinchao Chen, Pei Liu, Xinling Yang.

**Data curation:** Yatan Li, Wei Jia, Chen Chen, Cheng Chen, Jinchao Chen, Pei Liu, Xinling Yang.

**Formal analysis:** Yatan Li, Wei Jia, Chen Chen, Cheng Chen, Jinchao Chen, Pei Liu, Xinling Yang.

**Investigation:** Yatan Li, Chen Chen, Cheng Chen, Jinchao Chen, Pei Liu, Xinling Yang.

**Methodology:** Yatan Li, Wei Jia.

**Project administration:** Yatan Li, Wei Jia.

**Resources:** Yatan Li, Wei Jia.

**Software:** Yatan Li, Wei Jia.

**Supervision:** Yatan Li, Wei Jia.

**Validation:** Yatan Li, Wei Jia.

**Visualization:** Yatan Li, Wei Jia.

**Writing – original draft:** Yatan Li.

**Writing – review & editing:** Yatan Li, Wei Jia.

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
