## [Decision Letter · Decision Letter 0]

8 Dec 2024

PONE-D-24-22439Identification of biomarkers associated with inflammatory response in Parkinson's disease by bioinformatics and machine learningPLOS ONE

Dear Dr. Liu,

Thank you for submitting your manuscript to PLOS ONE. After careful consideration, we feel that it has merit but does not fully meet PLOS ONE’s publication criteria as it currently stands. Therefore, we invite you to submit a revised version of the manuscript that addresses the points raised during the review process.

We look forward to receiving your revised manuscript.

Kind regards,

Naseem Ahamad

Academic Editor

PLOS ONE

Journal Requirements:

3. Thank you for stating the following financial disclosure: [Project about Parkinson's disease peripheral neuropathy drug intervention fusion research innovation team (Item Number: 2023D14006).]. Please state what role the funders took in the study. If the funders had no role, please state: "The funders had no role in study design, data collection and analysis, decision to publish, or preparation of the manuscript." If this statement is not correct you must amend it as needed. Please include this amended Role of Funder statement in your cover letter; we will change the online submission form on your behalf.

4. Thank you for stating the following in the Acknowledgments Section of your manuscript: [Project about Parkinson's disease peripheral neuropathy drug intervention fusion research innovation team (Item Number: 2023D14006).] We note that you have provided funding information that is not currently declared in your Funding Statement. However, funding information should not appear in the Acknowledgments section or other areas of your manuscript. We will only publish funding information present in the Funding Statement section of the online submission form. Please remove any funding-related text from the manuscript and let us know how you would like to update your Funding Statement. Currently, your Funding Statement reads as follows: [Project about Parkinson's disease peripheral neuropathy drug intervention fusion research innovation team (Item Number: 2023D14006).] Please include your amended statements within your cover letter; we will change the online submission form on your behalf.

5. We note that your Data Availability Statement is currently as follows: [All relevant data are within the manuscript and its Supporting Information files.] Please confirm at this time whether or not your submission contains all raw data required to replicate the results of your study. Authors must share the “minimal data set” for their submission. PLOS defines the minimal data set to consist of the data required to replicate all study findings reported in the article, as well as related metadata and methods (https://journals.plos.org/plosone/s/data-availability#loc-minimal-data-set-definition). For example, authors should submit the following data: - The values behind the means, standard deviations and other measures reported; - The values used to build graphs; - The points extracted from images for analysis. Authors do not need to submit their entire data set if only a portion of the data was used in the reported study. If your submission does not contain these data, please either upload them as Supporting Information files or deposit them to a stable, public repository and provide us with the relevant URLs, DOIs, or accession numbers. For a list of recommended repositories, please see https://journals.plos.org/plosone/s/recommended-repositories. If there are ethical or legal restrictions on sharing a de-identified data set, please explain them in detail (e.g., data contain potentially sensitive information, data are owned by a third-party organization, etc.) and who has imposed them (e.g., an ethics committee). Please also provide contact information for a data access committee, ethics committee, or other institutional body to which data requests may be sent. If data are owned by a third party, please indicate how others may request data access.

6. PLOS requires an ORCID iD for the corresponding author in Editorial Manager on papers submitted after December 6th, 2016. Please ensure that you have an ORCID iD and that it is validated in Editorial Manager. To do this, go to ‘Update my Information’ (in the upper left-hand corner of the main menu), and click on the Fetch/Validate link next to the ORCID field. This will take you to the ORCID site and allow you to create a new iD or authenticate a pre-existing iD in Editorial Manager.

Reviewers' comments:

Reviewer's Responses to Questions

**Comments to the Author**

1. Is the manuscript technically sound, and do the data support the conclusions?

Reviewer #1: Yes

Reviewer #2: Yes

2. Has the statistical analysis been performed appropriately and rigorously? 

Reviewer #1: Yes

Reviewer #2: Yes

3. Have the authors made all data underlying the findings in their manuscript fully available?

Reviewer #1: Yes

Reviewer #2: Yes

4. Is the manuscript presented in an intelligible fashion and written in standard English?

Reviewer #1: Yes

Reviewer #2: Yes

5. Review Comments to the Author

Reviewer #1: The manuscript presents a promising approach to identifying biomarkers for Parkinson's Disease (PD) using bioinformatics and machine learning. The authors leverage gene expression data from the GEO database to find differentially expressed genes (DEGs) associated with the inflammatory response in PD. They then employ machine learning to identify candidate biomarker genes for further validation. Additionally, the use of single-cell transcriptomic analysis to explore candidate gene expression in different cell types strengthens the rationale. However, a few key areas require further clarification and improvement.

Specific Comments:

1. Figure Quality:

The quality of the figures needs significant improvement. The current resolution makes it difficult to evaluate the data and its relationship to the presented results. Ensure figures are clear, well-labeled, and have sufficient resolution for publication.

2. Validation Analysis:

The evaluation using the mRNA-miRNA network and drug sensitivity analysis requires a more elaborate discussion of the results. This includes:

A detailed explanation of the findings related to observed negative correlation with drugs. How negative correlation can help in finding the drug to putative drug targets.

Clarification on the rationale for considering weak correlations (Cor < 0.5) as validating evidence for drug sensitivity.

Discuss about relationship between the mRNA-miRNA network and putative pathways, whether the 17 miRNAs link to the discussed pathways.

3. Single-Cell Transcriptomic Analysis:

The discussion of the single-cell transcriptomic data needs further elaboration on the following aspects:

Specificity of expression of the candidate biomarker genes across different cell types.

Differential expression patterns observed in specific cell populations like neurons or glial cells.

Potential drug targets identified through drug sensitivity analysis that could be effective for these cell types.

By addressing these concerns, the authors can strengthen the manuscript and enhance its clarity and impact.

Reviewer #2: Reviewer Comments

No. Author submission Changes needed

1 Introduction: PD has a set of well-defined features, such as bradykinesia with resting tremor, as well as nonmotor symptoms including autonomic dysfunction, incontinence, and cognitive impairme[4].

*Changes need in Impairment

2 Fig 1. is not clear *

Image need in Higher definition

3 Provided all images in low definition, looks blurred

*Change into High definition with resolution more than 300dpi, follow the author instruction for preparing the Figures

4 All images provided without figure legends

*Provide legends on the top or bottom of the Figures

5 The author not followed the author instructions

*No line numbers mentioned. Use continuous line numbers

6 The author not followed reference style of the journal

*Follow the author instructions and reformat the references

7 Finally, the three biomarkers were found to be closely associated with excitatory neurons and microglia in single-cell transcriptomic analysis. In conclusion, IL18R1, NMUR1 and RELA could be potential therapeutic targets for PD in inflammatory response and new biomarkers for PD diagnosis.

*if possible show the drug target or biomarker interactions. Also show how the new biomarkers reacts with genes and show the response on inflammation.

6. PLOS authors have the option to publish the peer review history of their article (what does this mean? ). If published, this will include your full peer review and any attached files.

**Do you want your identity to be public for this peer review?** For information about this choice, including consent withdrawal, please see our Privacy Policy .

Reviewer #1: **Yes: ** PK

Reviewer #2: **Yes: ** Kannan Kanthaiah

---

## [Author Response · Author response to Decision Letter 0]

20 Dec 2024

Reviewer #1:

1.Is the manuscript technically sound, and do the data support the conclusions?

Reviewer #1:yes

Response-1:Thank you very much for recognizing our research.

2.Has the statistical analysis been performed appropriately and rigorously?

Reviewer #1:yes

Response-2:Thank you very much for recognizing our research.

3.Have the authors made all data underlying the findings in their manuscript fully available?

Reviewer #1:yes

Response-3:Thank you very much for recognizing our research.

4.Is the manuscript presented in an intelligible fashion and written in standard English?

Reviewer #1:yes

Response-4:Thank you very much for recognizing our research.

5.The quality of the figures needs significant improvement. The current resolution makes it difficult to evaluate the data and its relationship to the presented results. Ensure figures are clear, well-labeled, and have sufficient resolution for publication.

Response-5:Thank you for your valuable comments. We attach great importance to the quality of the graphs and have re-exported the images in tif format at dpi=300 resolution as per the requirements of the journal.

6.Validation Analysis:

The evaluation using the mRNA-miRNA network and drug sensitivity analysis requires a more elaborate discussion of the results. This includes:

(1)A detailed explanation of the findings related to observed negative correlation with drugs. How negative correlation can help in finding the drug to putative drug targets.

(2)Clarification on the rationale for considering weak correlations (Cor < 0.5) as validating evidence for drug sensitivity.

(3)Discuss about relationship between the mRNA-miRNA network and putative pathways, whether the 17 miRNAs link to the discussed pathways.

Response-6:

Thank you for your valuable comments. Below is a detailed response to the above questions. In addition, we have added detailed descriptions to the manuscript in the appropriate places.

(1) Negative correlation helps to find potential drug targets of the drugs. The results showed that IL18R1 was negatively correlated with the drug BP-1-102 (Cor = -0.36, P = 0.004).RELA was negatively correlated with the drugs ARRY-162 (Cor = -0.37, P = 0.004) and Tamoxifen (Cor = -0.3, P = 0.018). A negative correlation means that when the expression level of the gene increases, the effect of the drug tends to diminish. Xu, F., et al. [1] found a negative correlation between camptothecin drugs and SLC1A5 expression in a drug sensitivity analysis of SLC1A5 expression with classical chemotherapeutic drugs.

(2) As this study is in the pre-exploratory stage, these weak correlations are still informative and may reveal potentially complex biological mechanisms. In addition, the relationship between genes and drug responses is usually nonlinear, so a single weak correlation does not imply a lack of biological significance. For example, Song, D., et al [2], in exploring the sensitivity of small molecule drugs associated with NRAV expression, found that the drug Sapitinib, which is weakly associated with NRAV expression, induced apoptosis and inhibited phosphorylated epidermal growth factor receptor and its downstream pathway.

(3) miRNAs are associated with the pathways discussed. In our study, we mainly focused on mRNA biomarkers, so the pathway analysis was performed based on mRNAs. Based on this, we mainly explored the relationship between mRNAs and their associated pathways and mined potential miRNA associations through mRNAs. However, based on your suggestion, we further explored the role of miRNA target genes in the relevant pathways (S1 Fig.tif). Through KEGG and GO analysis of miRNA target genes, we found that the pathways of miRNA target genes overlapped with the pathways we identified at the mRNA level, specifically the virus process and ECM-receptor interaction pathways. These pathways, although significant (P<0.05), were ranked low and therefore did not appear in the pathway heatmap for mRNAs. This result further supports the possible interaction of mRNA-miRNA networks in relevant biological processes.

S1 Fig KEGG and GO analysis of miRNA target genes

7. Single-Cell Transcriptomic Analysis:

The discussion of the single-cell transcriptomic data needs further elaboration on the following aspects:

(1)Specificity of expression of the candidate biomarker genes across different cell types.

(2)Differential expression patterns observed in specific cell populations like neurons or glial cells.

(3)Potential drug targets identified through drug sensitivity analysis that could be effective for these cell types.

By addressing these concerns, the authors can strengthen the manuscript and enhance its clarity and impact.

Response-7:

Thank you for your valuable comments. Below is a detailed description for the specific expression of IL18R1, NMUR1 and RELA in different cell types. In addition, we have also added a detailed description in the discussion section of Chapter 4 of the manuscript.

(1) IL18R1 is specifically expressed in T cells and Endothelial cells. It suggests that IL18R1 may play an important role in the immune response, especially in T cells. T cells are key cells of the immune system and are involved in recognizing and attacking infected or tumor cells[3]. Whereas IL18R1 is a subunit of the IL-18 receptor, IL-18 is a pro-inflammatory cytokine that is commonly associated with immune response and inflammation[4]. The specific expression of IL18R1 in T cells may be related to its role in the immune response[5]. RELA is expressed in almost all cell populations.RELA is part of the NF-κB family, and NF-κB plays an important role in many biological processes, including immune response, inflammatory response, cell survival and proliferation[6]. The expression of RELA in virtually all cell populations suggests that it is a fundamental and broadly regulated factor that may maintain essential functions and responsiveness in the physiological state of the cell, especially in response to inflammation, infection, or cellular damage[7]. NMUR1 is specifically expressed in T cells.NMUR1 is usually associated with neuropeptide signaling, which may influence the immune response by regulating T cell activation, proliferation, or function[8].

(2) In single-cell transcriptome analysis, IL18R1 and NMUR1 were found to have a small amount of expression in excitatory neurons (ExN), inhibitory neurons (InN) cells, but more often found to have a high specific expression in T cells and Endothelial cells. While RELA has high expression in excitatory neurons (ExN), inhibitory neurons (InN) cells.

(3) Thank you very much for your valuable questions. Our study focuses on screening biomarker genes by transcriptomics and verifying the specific expression of these markers in different cell types using single-cell transcriptomics, aiming to reveal their distribution and expression characteristics at the cellular level. Regarding drug sensitivity analysis, we have indeed performed preliminary correlation analysis at the level of transcriptomics data to identify potential drug targets. However, drug sensitivity analysis was not carried out on single-cell transcriptomics data as the main goal of this study was to validate the specific expression of the marker genes in different cell types rather than to explore the direct effects of the drugs on these cell types. However, in the future we will focus on drug sensitivity analysis on single-cell transcriptomics data.

Reviewer #2:

1.Is the manuscript technically sound, and do the data support the conclusions?

Reviewer #2:yes

Response-1: Thank you very much for recognizing our research.

2. Has the statistical analysis been performed appropriately and rigorously?

Reviewer #2:yes

Response-2: Thank you very much for recognizing our research.

3.Have the authors made all data underlying the findings in their manuscript fully available?

Reviewer #2:yes

Response-3: Thank you very much for recognizing our research.

4.Is the manuscript presented in an intelligible fashion and written in standard English?

Reviewer #2:yes

Response-4: Thank you very much for recognizing our research.

5.Introduction: PD has a set of well-defined features, such as bradykinesia with resting tremor, as well as nonmotor symptoms including autonomic dysfunction, incontinence, and cognitive impairme

*Changes need in Impairment

Response-5: Thank you for your valuable comments. We have amended the paragraph.

6.Fig 1. is not clear *

Image need in Higher definition

Response-6: Thank you for your valuable comments. We attach great importance to the quality of the graphs and have re-exported the images in tif format at dpi=300 resolution as per the requirements of the journal. And uploaded separately in the system.

7.Provided all images in low definition, looks blurred

*Change into High definition with resolution more than 300dpi, follow the author instruction for preparing the Figures

Response-7: Thank you for your valuable comments. We have re-exported the image to a resolution of 300 dpi as required by the journal and set the image format to tif and uploaded it separately in the system.

8.All images provided without figure legends

*Provide legends on the top or bottom of the Figures

Response-8: Thank you for your valuable comments. We have provided legends to the right or left of the images, for example, in Figure 2A, Parkinson's patients are abbreviated as PD and the corresponding legend is in blue. Except for that, images were uploaded individually as required by the journal. Figure captions for each figure are placed in the body of the manuscript.

9.The author not followed the author instructions

*No line numbers mentioned. Use continuous line numbers

Response-9: Thank you for your valuable comments. We have added line numbers and page numbers to the manuscript as required by the journal manuscript format.

10.The author not followed reference style of the journal

*Follow the author instructions and reformat the references

Response-10: Thank you for your comments. We have updated the reference format and changed it to the “Vancouver” style required by the journal.

11.Finally, the three biomarkers were found to be closely associated with excitatory neurons and microglia in single-cell transcriptomic analysis. In conclusion, IL18R1, NMUR1 and RELA could be potential therapeutic targets for PD in inflammatory response and new biomarkers for PD diagnosis.

*if possible show the drug target or biomarker interactions. Also show how the new biomarkers reacts with genes and show the response on inflammation.

Response-11: Thank you for your valuable comments. Below is a detailed response to your questions.

(1) Biomarker interactions: we have mapped the correlation heatmap of differentially expressed genes for inflammatory response in Section 3.1 of the article, which includes the correlation of the biomarkers screened, IL18R1, NMUR1 and RELA.

(2) Since our focus is on exploring the interactions between biomarkers and drugs, the analysis of drug-drug interactions is not addressed. In Section 3.7, the interpretation of interactions between drug targets and biomarkers is carried out. The sensitivities of three biomarkers, IL18R1, NMUR1 and RELA, corresponding to different drugs were detailed, where IL18R1 was positively associated with the Alisertib (Cor = 0.31, P = 0.016), Azacitidine (Cor = 0.33, P = 0.01) and Motesanib (Cor = 0.35, P = 0.007) drugs, IL18R1 was positively correlated with BP-1-102 (Cor = -0.36, P = 0.004) drug, NMUR1 was positively correlated with Lexibulin (Cor = 0.31, P = 0.017) and norvir (Cor = 0.48, P < 0.001) drugs, and RELA was positively correlated with CAMPTOTHECIN (Cor = 0.32, P = 0.012) and Clofarbine (Cor = 0.3, P = 0.019) drugs, and RELA was positively correlated with ARRY-162 (Cor = -0.37, P = 0.004) and Tamoxifen (Cor = -0.3, P = 0.018) and 7 other drugs were negatively correlated.

(3) Biomarkers associated with inflammatory response Explanation: a. The experiments were first performed by searching for candidate genes that might serve as biomarkers among the genes associated with the inflammatory response. b. These biomarkers were found to be associated with the differential expression of Mast cells and Neutrophils cells by immune cell infiltration analysis. c. The biomarkers were found to be associated with the differential expression of Mast cells and Neutrophils cells by the immune cell infiltration analysis. Mast cells and Neutrophils cells are key cells involved in immune responses and inflammatory processes[9, 10]. Therefore, the differential expression of markers with them may imply that these markers play a role in regulating the inflammatory response.

References

1. Xu, F., H. Wang, H. Pei, Z. Zhang, L. Liu, L. Tang, S. Wang, and B.-C. Ren, SLC1A5 prefers to play as an accomplice rather than an opponent in pancreatic adenocarcinoma. Frontiers in Cell and Developmental Biology, 2022. 10: p. 800925.

2. Song, D., X. Wang, Y. Wang, W. Liang, J. Luo, J. Zheng, and K. Zhu, Integrated analysis of N1-methyladenosine methylation regulators-related lncRNAs in hepatocellular carcinoma. Cancers, 2023. 15(6): p. 1800.

3. Ruf, B., T.F. Greten, and F. Korangy, Innate lymphoid cells and innate-like T cells in cancer—at the crossroads of innate and adaptive immunity. Nature Reviews Cancer, 2023. 23(6): p. 351-371.

4. Di, Y., Z. Wang, J. Xiao, X. Zhang, L. Ye, X. Wen, J. Qin, L. Lu, X. Wang, and W. He, ACSL6-activated IL-18R1–NF-κB promotes IL-18–mediated tumor immune evasion and tumor progression. Science Advances, 2024. 10(38): p. eadp0719.

5. Lutz, V., V.M. Hellmund, F.S. Picard, H. Raifer, T. Ruckenbrod, M. Klein, T. Bopp, R. Savai, P. Duewell, and C.U. Keber, IL18 receptor signaling regulates tumor-reactive CD8+ T-cell exhaustion via activation of the IL2/STAT5/mTOR pathway in a pancreatic cancer model. Cancer immunology research, 2023. 11(4): p. 421-434.

6. Guo, Q., Y. Jin, X. Chen, X. Ye, X. Shen, M. Lin, C. Zeng, T. Zhou, and J. Zhang, NF-κB in biology and targeted therapy: new insights and translational implications. Signal Transduction and Targeted Therapy, 2024. 9(1): p. 53.

7. Khan, A., Y. Zhang, N. Ma, J. Shi, and Y. Hou, NF-κB role on tumor proliferation, migration, invasion and immune escape. Cancer Gene Therapy, 2024: p. 1-12.

8. Ye, Y., Z. Liang, and L. Xue, Neuromedin U: potential roles in immunity and inflammation. Immunology, 2021. 162(1): p. 17-29.

9. Mencarelli, A., P. Bist, H.W. Choi, H.J. Khameneh, A. Mortellaro, and S.N. Abraham, Anaphylactic degranulation by mast cells requires the mobilization of inflammasome components. Nature Immunology, 2024. 25(4): p. 693-702.

10. Mihlan, M., S. Wissmann, A. Gavrilov, L. Kaltenbach, M. Britz, K. Franke, B. Hummel, A. Imle, R. Suzuki, and M. Stecher, Neutrophil trapping and nexocytosis, mast cell-mediated processes for inflammatory signal relay. Cell, 2024. 187(19): p. 5316-5335. e28.

---

## [Decision Letter · Decision Letter 1]

28 Jan 2025

PONE-D-24-22439R1Identification of biomarkers associated with inflammatory response in Parkinson's disease by bioinformatics and machine learningPLOS ONE

Dear Dr. Liu,

Thank you for submitting your manuscript to PLOS ONE. After careful consideration, we feel that it has merit but does not fully meet PLOS ONE’s publication criteria as it currently stands. Therefore, we invite you to submit a revised version of the manuscript that addresses the points raised during the review process.

We look forward to receiving your revised manuscript.

Kind regards,

Naseem Ahamad

Academic Editor

PLOS ONE

Reviewers' comments:

Reviewer's Responses to Questions

**Comments to the Author**

1. If the authors have adequately addressed your comments raised in a previous round of review and you feel that this manuscript is now acceptable for publication, you may indicate that here to bypass the “Comments to the Author” section, enter your conflict of interest statement in the “Confidential to Editor” section, and submit your "Accept" recommendation.

Reviewer #1: All comments have been addressed

Reviewer #2: All comments have been addressed

2. Is the manuscript technically sound, and do the data support the conclusions?

Reviewer #1: Yes

Reviewer #2: Yes

3. Has the statistical analysis been performed appropriately and rigorously? 

Reviewer #1: Yes

Reviewer #2: Yes

4. Have the authors made all data underlying the findings in their manuscript fully available?

Reviewer #1: Yes

Reviewer #2: Yes

5. Is the manuscript presented in an intelligible fashion and written in standard English?

Reviewer #1: Yes

Reviewer #2: Yes

6. Review Comments to the Author

Reviewer #1: I am pleased with the justifications and improvements made to the manuscript. However, I have noticed that some of the provided justifications have not been incorporated into the manuscript text. I recommend updating the discussion section to reflect these justifications for greater clarity and alignment with the revisions.

Reviewer #2: Even though the author answered all the questions raised by the reviewer, still the paper is at its best. Again, the author given All the images in lo w resolution. Kindly provide with high definition as mentioned in Author instruction.

7. PLOS authors have the option to publish the peer review history of their article (what does this mean? ). If published, this will include your full peer review and any attached files.

**Do you want your identity to be public for this peer review?** For information about this choice, including consent withdrawal, please see our Privacy Policy .

Reviewer #1: **Yes: ** Prince Kumar

Reviewer #2: **Yes: ** Kannan

---

## [Author Response · Author response to Decision Letter 1]

31 Jan 2025

Reviewer #1:

1. I am pleased with the justifications and improvements made to the manuscript. However, I have noticed that some of the provided justifications have not been incorporated into the manuscript text. I recommend updating the discussion section to reflect these justifications for greater clarity and alignment with the revisions.

Response-1: Thank you so much for your positive feedback and valuable suggestions on our manuscript. We have updated Chapter 4 (Discussion) and Chapter 5 (Conclusion) of the manuscript to ensure that all the explanations we provided in our responses are fully integrated into the main text. Additionally, in response to your question, "Discuss the relationship between the mRNA-miRNA network and the putative pathways, and whether the 17 miRNAs are related to the discussed pathways," we have provided a detailed description in Section 3.8.

Reviewer #2:

1.Again, the author given All the images in lo w resolution. Kindly provide with high definition as mentioned in Author instruction.

Response-1: Thank you for your attention to the issue of image resolution. We have uploaded the images in TIF format with a resolution of 300 dpi, as required by the journal. However, we have noticed that the submission system compresses the images when generating the PDF, which results in a reduction of image clarity. This compression is an automatic process that we cannot control. We assure you that the original images we uploaded meet the journal’s high-resolution requirements. You can download the original images separately through the submission system, or you can click on the links to the original images within the PDF to view them in high resolution.

---

## [Decision Letter · Decision Letter 2]

16 Feb 2025

Identification of biomarkers associated with inflammatory response in Parkinson's disease by bioinformatics and machine learning

PONE-D-24-22439R2

Dear Dr. Liu,

We’re pleased to inform you that your manuscript has been judged scientifically suitable for publication and will be formally accepted for publication once it meets all outstanding technical requirements.

Kind regards,

Naseem Ahamad

Academic Editor

PLOS ONE

Additional Editor Comments (optional):

Reviewers' comments:

Reviewer's Responses to Questions

**Comments to the Author**

1. If the authors have adequately addressed your comments raised in a previous round of review and you feel that this manuscript is now acceptable for publication, you may indicate that here to bypass the “Comments to the Author” section, enter your conflict of interest statement in the “Confidential to Editor” section, and submit your "Accept" recommendation.

Reviewer #1: All comments have been addressed

Reviewer #2: All comments have been addressed

2. Is the manuscript technically sound, and do the data support the conclusions?

Reviewer #1: Yes

Reviewer #2: Yes

3. Has the statistical analysis been performed appropriately and rigorously? 

Reviewer #1: Yes

Reviewer #2: Yes

4. Have the authors made all data underlying the findings in their manuscript fully available?

Reviewer #1: Yes

Reviewer #2: Yes

5. Is the manuscript presented in an intelligible fashion and written in standard English?

Reviewer #1: Yes

Reviewer #2: Yes

6. Review Comments to the Author

Reviewer #1: (No Response)

Reviewer #2: The author address all the required comment given by me hence I recommend the article to accept. Kindly go with the decision of other reviewer also.

7. PLOS authors have the option to publish the peer review history of their article (what does this mean? ). If published, this will include your full peer review and any attached files.

**Do you want your identity to be public for this peer review?** For information about this choice, including consent withdrawal, please see our Privacy Policy .

Reviewer #1: **Yes: ** Prince Kumar

Reviewer #2: **Yes: ** Kannan Kanthaiah

---

## [Editor Report · Acceptance letter]

PONE-D-24-22439R2

PLOS ONE

Dear Dr. Liu,

I'm pleased to inform you that your manuscript has been deemed suitable for publication in PLOS ONE. Congratulations! Your manuscript is now being handed over to our production team.

Kind regards,

on behalf of

Dr. Naseem Ahamad

Academic Editor

PLOS ONE